# The Ability of Shiga Toxin-Producing *Escherichia coli* to Grow in Raw Cow’s Milk Stored at Low Temperatures

**DOI:** 10.3390/foods11213411

**Published:** 2022-10-28

**Authors:** Lene Idland, Erik G. Bø-Granquist, Marina Aspholm, Toril Lindbäck

**Affiliations:** 1Department of Paraclinical Sciences, Faculty of Veterinary Medicine, Norwegian University of Life Sciences, 1432 Ås, Norway; 2Department of Production Animal Clinical Sciences, Faculty of Veterinary Medicine, Norwegian University of Life Sciences, 1432 Ås, Norway

**Keywords:** Shiga toxin-producing *Escherichia coli*, raw cow’s milk, unpasteurized, storage, temperature, food safety, Shiga-toxin, bacteriophage

## Abstract

Despite the lack of scientific evidence, some consumers assert that raw milk is a natural food with nutritional and immunological properties superior to pasteurized milk. This has led to the increased popularity of unpasteurized cow milk (UPM) and disregard for the risks of being exposed to zoonotic infections. Dairy cattle are healthy carriers of Shiga toxin (Stx)-producing *E. coli* (STEC), and contaminated UPM has caused STEC outbreaks worldwide. The association between STEC, carrying the *eae* (*E. coli* attachment effacement) gene, and severe diseases is well-established. We have previously isolated four *eae* positive STEC isolates from two neighboring dairy farms in the Southeast of Norway. A whole genome analysis revealed that isolates from different farms exhibited nearly identical genetic profiles. To explore the risks associated with drinking UPM, we examined the ability of the isolates to produce Stx and their growth in UPM at different temperatures. All the isolates produced Stx and one of the isolates was able to propagate in UPM at 8 °C (*p* < 0.02). Altogether, these results highlight the risk for STEC infections associated with the consumption of UPM.

## 1. Introduction

Enterohemorrhagic *Escherichia coli* (EHEC) is a globally distributed intestinal pathogen associated with human diarrhea, hemorrhagic colitis, and hemolytic uremic syndrome (HUS) [1]. The term “EHEC” is restricted to Shiga toxin-producing *E. coli* (STEC) associated with human disease. The main reservoir of STEC is the ruminant digestive tract and undercooked beef and unpasteurized milk are considered high-risk foods for STEC infections [1,2]. In 2020, 4446 cases of EHEC disease and 13 deaths were reported in the EU [3]. The first large outbreak of EHEC occurred in the USA in 1982 and was caused by a strain of serotype O157:H7 [4]. Since then, other serotypes have also been associated with outbreaks of EHEC disease [5,6,7]. The most known non-O157:H7 strain is O104:H4, which caused 855 cases of HUS and 50 fatalities during a large European outbreak in 2011 [2]. EHEC has a low infectious dose of 10–100 colony-forming units [8,9], and insufficient food decontamination practices increases the risk for EHEC infections.

STEC can produce two different types of Shiga toxin, Stx1 and Stx2, both comprising several subtypes. Stx2 is more often associated with HUS than Stx1, and Stx2a is considered as the most potent subtype of the toxin [2]. The Stx-encoding genes are carried by temperate bacteriophages [2], and the pathogenic potential of STEC has been suggested to be influenced by the “EHEC phage replication unit” (Eru) located in the phage genome [10,11]. The life cycle of temperate phages is regulated by the CI repressor protein, which represses the transcription of the replication proteins during the lysogenic state of the phage [12,13]. The de-repression of CI results in the production of Stx and new phage particles [14]. Based on similarities in its amino acid sequence, the CI protein of Stx phages has been grouped into eight major clades (I–VIII) [11]. Exactly how the variability in the CI sequence influences its regulatory properties and potentially the virulence properties of its host STEC strain have not been explored so far.

Stx production combined with the ability to adhere to the intestinal epithelium via the adhesion protein intimin are believed to be necessary for STEC to cause severe disease. The intimin-encoding gene *(eae)* is part of the locus of the enterocyte effacement pathogenicity island (LEE-PAI), which encodes proteins responsible for introducing attaching and effacing (A/E) lesions to the epithelial cells [15]. Similar to CI, intimins display a structural diversity that potentially reflects differences in host cell tropism. The most common types of intimin are α, β, γ, ε, ζ, and η [15]. The β-type has been shown to predominate in non-O157 STEC strains from diarrheal patients, while cattle isolates more often carry the ζ-type [16]. The presence of *eae* is associated with a higher risk of developing HUS [17].

EHEC is regarded as an emerging public health challenge as new pathotypes and serotypes constantly appear [18,19,20]. Milk contaminated with pathogens causes foodborne disease worldwide, and 33% of all reported milk-borne disease outbreaks in England and Wales between 1992–2000 were caused by EHEC [21]. Previous studies have shown that 27, 13, and 5% of cattle from Portugal [22], US [23], and the EU [3] carry STEC, respectively. A study from Finland showed that 2% of on-farm, in-line milk filters were positive for STEC of the serotype O157:H7 [24], while in Norway, STEC has been detected in 7% of milk filters [25]. As STECs are carried by asymptomatic cows and frequently occur in dairy farm environments [26], the milk from these sources can easily be contaminated during the milking process. The lack of effective preventive measures in the primary production of milk makes pasteurization necessary to ensure food safety. Pasteurization at 72 °C for 15 s has shown to be very effective for the inactivation of STEC [27].

Low-temperature storage is important for preventing microbial growth in milk [28]. Previous studies have shown that STEC is not able to grow at 4 °C, but proliferation has been observed at inadequate refrigeration temperatures [29,30]. It has been shown that *E. coli* of the serotype O157:H7 grows in unpasteurized and pasteurized milk with a 2- to 3-log CFU/mL increase at 8 °C within a time period of seven days [31]. The European Food Safety Authority (EFSA) recommend that certain unpasteurized and low-pasteurized dairy products should be stored below 5 °C to minimize microbial growth [32]. However, the temperature in domestic refrigerators has been shown to vary between 7.0 ± 2.7 °C and 6.1 ± 2.8 °C for southern and northern European countries, respectively [33]. In addition, short breaks in the cold chain, for example, during meals, represent an additional but unexplored factor that may add to the risk of consuming UPM.

To further assess the food safety risk associated with the consumption of UPM, we need to gain more knowledge on the genetic- and growth characteristics of the STECs isolated from raw cows’ milk. In the present study, we have compared the genome of four STECs isolated from milk [25] with a focus on their content of virulence-associated genes and Stx phages. The isolates were tested for their survival and growth in UPM milk, incubated at recommended and abused storage temperatures, and for the production of Stx at the body temperature of a human host. Altogether, the results highlight the risk for EHEC infections associated with the consumption of UPM, particularly if the milk has been stored at an abused temperature.

## 2. Materials and Methods

### 2.1. Culturing Conditions

This study comprises four *stx*- and *eae*-positive *E. coli* isolates from Norwegian dairy farms [25]. Three of the isolates were from the same farm, two from fecal samples (S2 and S4) and one from an in-line milk filter sample (S3), while the fourth isolate was isolated from a fecal sample (S1) at a nearby farm. The isolates were collected at two different sampling occasions separated by five months (Table 1). Raw milk, from the dairy cattle breed Norwegian Red, was collected from a bulk tank at the Center for Livestock Experiments at the Norwegian University of Life Sciences and used as cultivation medium in the growth experiments. The milk was collected in batches of approximately 2 L at two different occasions (September 2021 and April 2022) and aliquoted in 40 mL batches in Falcon tubes and frozen at −20 °C until use.

To explore the ability of the STEC isolates to grow at different temperatures, over-night cultures of the respective isolates grown in Lysogeny broth (LB) were diluted to OD_600_ = 0.3, whereof 0.5 µL were transferred to 40 mL of thawed raw milk. Immediately after inoculation, 10 µL of the milk samples was plated on CHROMagar^TM^ STEC (Kanto Chemical Co., Tokyo, Japan) and incubated at 37 °C for 24 h to enumerate the start concentration of STEC. The inoculated raw milk samples were then incubated at five different temperature settings: optimal refrigerator temperature (4 °C), abused refrigerator temperatures (6 °C and 8 °C), room temperature (20 °C), and a temperature setting mimicking the situation when milk is kept at room temperature during meals (4 °C except for 1.5 h daily at 20 °C). To determine the temperature fluctuation of the samples incubated this way, the temperature was recorded in an uninoculated 40 mL raw milk sample every 15 min during the 20 °C incubation and until the milk temperature had returned to 4 °C, which encompassed a total time of 4.5 h. For enumeration of STEC in the raw milk samples incubated at different temperatures, dilutions of the samples were plated on CHROMagar^TM^ STEC agar after 24, 48, and 72 h of incubation. The growth ratio, used as indicator of growth, was calculated by dividing the number of STEC colonies appearing on the plates after 24, 48, and 72 h by the number of the STEC colonies present in the cultures at time zero.

To determine growth of the STEC isolates in laboratory media without the impact of competing bacteria, each isolate was inoculated into 40 mL LB and incubated at 20 °C. For enumeration, appropriate dilutions of the cultures were plated on LB agar after 0 and 24 h. All experiments were performed in three biological replicates, except for STEC incubated in raw milk at 20 °C, which was only performed with two replicates. To exclude the presence of STEC in the two raw milk batches used, 6 × 100 µL raw milk samples from each batch were plated on CHROMagar^TM^ STEC agar and incubated according to manufacturer’s instructions.

### 2.2. Stx Production

A volume of 100 µL overnight LB-cultures was transferred to 5 mL fresh LB and incubated at 37 °C with agitation at 250 rpm until the optical density reached 0.5 at 600 nm (OD_600_). Half of these cultures were induced by addition of 0.5 µg/mL of Mitomycin C (MMC). Both induced and uninduced cultures were incubated further for 3 h. Six samples, three induced and thee uninduced, were processed and analyzed with respect to Stx content for each STEC isolate. The Stx content was measured in 1:20 dilutions of the cultures using the semi-quantitative enzyme immunoassay RIDASCREEN^®^ Verotoxin kit (R-biopharm, Darmstadt, Germany) according to the manufacturer.

### 2.3. Genome Sequence Analyses

DNA for long-read sequencing was extracted using Nanobind CBB Big DNA Kit (NB-900-001-01, Circulomics, Baltimore, MD, USA), according to the manufacturer’s instructions (Nanobind HMW DNA Extraction protocol for Gram-Negative Bacteria, 2021). Oxford Nanopore Technologies’ “Ligation Sequencing kit” (SQK-LSK109, Oxford Nanopore Technologies Plc., Oxford, UK) was used for library preparation and “Native Barcoading Expansions” 1–12 (EXP-NBD104, Oxford Nanopore Technologies Plc., Oxford, UK) for barcoding the libraries. Nanopore sequencing was performed on a FLO-Min106 (R9.4.1, Oxford Nanopore Technologies Plc., Oxford, UK) flow cell. Recovered reads were assembled using the Flye assembler implemented in the “Dragonflye”-pipeline (https://github.com/rpetit3/dragonflye, v.1.0.12 (accessed on 25 March 2022)), which also performs adapter removal and assembly polishing. Virulence and antimicrobial resistance genes, core genome MLST type, and serotype were identified using the following tools on the CGE website: VirulenceFinder 2.0 [34,35], ResFinder 4.1 [36,37,38], cgMLSTFinder 1.1 [39,40], and SerotypeFinder 2.0 [41]. Prophage sequences were identified and annotated using the Phaster web software [42]. Isolate diversities were examined by SNP using Snippy v. 4.6.01 (https://github.com/tseemann/snippy (accessed on 20 May 2022)) and Mauve v2.4.0 (https://darlinglab.org/mauve/mauve.html (accessed on 5 May 2022)) were used to align the genomes (default parameters). This Whole Genome Shotgun project has been deposited at DDBJ/ENA/GenBank under the accession JANWGC000000000 to JANWGF000000000 (Table 1).

### 2.4. Statistics

For all growth experiments, a two-tailed paired Student’s *t*-test, performed via Microsoft Office Excel, was used to test for statistically significant differences between average CFU determined at two different time points. *p*-values equal to or below 0.05 were considered significant. Standard deviation was calculated using Excel.

## 3. Results

### 3.1. Genetic Characterization of STEC Isolates from Raw Milk

Three of the four STEC isolates included in this study originated from the same farm (S2, S3, and S4); two were collected from fecal samples (S2 and S4) and one from an in-line milk filter sample (S3). Isolate S2 was collected five months prior to S3 and S4. The fourth isolate (S1) originates from a fecal sample from a second farm located within 10 km from farm one. The characteristics of the four STECs are listed in Table 1.

A genome sequence analysis revealed that isolates S1, S3, and S4 are highly similar and differ by only 19–23 SNPs, suggesting that these isolates are clonal (Figure 1).

S1, S3, and S4 exhibit 5.2 Mb chromosomes and the sequence analysis shows that they are of the serotype ONT:H28 and that they belong to the core genome multi-locus sequence type (cgMLST) 7679. Their genomes harbor the LEE-PAI-encoding intimin gamma (*eae*) and the gene encoding the translocated intimin receptor (Tir). The LEE-PAI is 99% identical over 33.3 kbp to the *E. coli* O157:H7 strain EDL933 (NCBI accession number NZ_CP008957) from the US outbreak in 1982 [4]. The lambdoid Stx1 phage of isolates S1, S3, and S4 is 99% identical over 22.8 kbp to Phage BP-4795 (*E. coli,* strain 4795/97, serotype O84:H4 human, Germany 1997) [15,43]. The CI repressor of this phage belongs to Clade V [11]. All three isolates carry a circular plasmid of 55 kbp encoding a heat-stable toxin (*astA*) and enterohaemolysin (*ehxA*) [44,45]. The heat-stable toxin is known to cause sporadic diarrhea in humans and animals [46], while enterohaemolysin is associated with bloody diarrhea and HUS [47]. Furthermore, in the genome of each isolate, a total of 18 prophages of varying completeness were identified by Phaster [42,48]. The Stx phage harbored by these stains is of the lambdoid type and encodes Stx1a [4,49].

The genome of isolate S2 is highly different from those of S1, S3, and S4 (Figure 1). It comprises two circular contigs including a chromosome of 5.4 Mbp and a plasmid of 80 kbp. A DNA-typing analysis revealed that the isolate belongs to serotypes O108:H25 and cgMLST 141324. S2 carries a bacteriophage of Eru type 1 and a CI repressor belonging to Clade II [10,11]. The phage encodes the Stx2a type of Stx [50] and shares 99% identity over 22.2 kbp covering the replication region of the Stx2 phage TL-2011c (NCBI accession number NC_019442), which was carried by a highly virulent EHEC strain that caused an outbreak in Norway in 2006 [51].

Similar to the other three isolates, S2 harbors LEE-PAI including both *eae* and *tir*. The DNA sequence of the five LEE operons shows 87% identity over 30 kbp to the corresponding sequence of *E. coli* O157:H7 strain EDL933 (NCBI accession number NZ_CP008957). The 80 kbp plasmid of isolate S2 contains both *astA* and *ehxA*. Phaster identified 30 prophage regions on the chromosome and one prophage on the plasmid in isolate S2. ResFinder 4.1 did not detect antimicrobial resistance genes in any of the four isolates.

### 3.2. Stx Production

To explore the virulence potential of the STEC isolates, the Stx production was examined during growth in LB at 37 °C, with and without induction by MMC. All four isolates produced Stx and the levels were higher three hours post-induction with MMC compared to the uninduced samples (Figure 2).

### 3.3. Growth Characteristics of STEC Isolates in Raw Milk at Different Storage Temperatures

To examine the ability of the four STEC isolates to survive and grow in UPM, 40 mL raw milk samples were inoculated with approximately 3000–5000 CFU/mL of STEC culture. The samples were then incubated at 4 °C, 6 °C, 8 °C, and 20 °C for 72 h. After 0, 24, and 72 h, the samples were plated on Chromagar^TM^ STEC for enumeration. The growth ratios were calculated by dividing the number of STEC at 24 and 72 h by the number of bacteria inoculated into the milk.

At 4 °C, an average reduction in CFU (growth ratio below 1) was observed for all four isolates after storage for 24 h. The reduction was, however, not significant for any of the four isolates (Figure 3a). For isolates S1, S2, and S3, the number of CFU was further reduced over the next 48 h, while the level of isolate S4 remained unchanged (Figure 3a). The reduction in bacterial levels seen after 72 h, compared to the levels at the start of cultivation, was only significant for isolate S3 (*p* < 0.01). At 6 °C, a decrease in CFU/mL was observed during the first 24 h (*p* ≤ 0.05 for isolate S1 and S4) but the cell death stopped after 24 h (Figure 3b). At 8 °C, S1, S3, and S4 multiplied over the first 24 h of storage (growth ratio above 1), and all strains showed increased CFU counts after 72 h (Figure 3c). The increase in CFU/mL after 72 h of storage, compared to the CFU at the start of cultivation, was significant only for isolate S1 (*p* < 0.02). There was a large difference in growth between isolate S2 and the three other isolates at 20 °C (Figure 3d).

Under abused conditions, wherein the inoculated milk samples were kept at 4 °C but exposed to 20 °C for 90 min every 24 h, a trend of positive growth ratios was observed after 72 h of storage. However, only the increase in CFU/mL between 24 h and 48 h (*p* < 0.01) and between 24 h and 72 h (*p* < 0.05) for isolate S2 were significant (Figure 4a). The average growth ratios were lower than those observed at 8 °C (Figure 3c). The growth ratios of the four isolates inoculated into LB and incubated for 24 h indicate that the ability to grow in LB at 20 °C is similar for the four isolates (Figure 4b), and that they multiply faster in LB compared to unpasteurized milk at 20 °C.

Recordings of the temperature fluctuation in the 40 mL raw milk showed that after reaching 20 °C, it took >3 h for the milk to reach below 5 °C (Figure 5).

## 4. Discussion

Cattle represent a reservoir of STEC, and the consumption of unpasteurized milk is, therefore, considered an important risk factor for contracting milk-borne STEC infections [1,2]. Herein, we explore the pathogenic potential of four *eae*-positive STECs (S1–S4) isolated from Norwegian dairy herds and their ability to grow in UPM stored under optimal and abused temperature conditions.

The genome analysis showed that isolates S1, S3, and S4 are clonal even though they were isolated from two different farms and S1 was isolated seven months prior to S3 and S4. This indicates that STEC has been transmitted between the two farms and persisted in the farm environment over time. Previous studies have shown that *E. coli* O157:H7 can survive for 99 days in soil [52] and 13 weeks in lake water at 15 °C [53]. The clonal isolates S4 from feces and S3 from a milk filter were isolated the same day from the same farm, which strongly suggests that STEC can be transmitted from feces to the raw milk.

To explore the potential of the four isolates to cause disease, the genomes of the isolates were examined with respect to known virulence-associated genes. Isolates S1, S3, and S4 carry genes encoding Stx1a, while isolate S2 carries genes encoding Stx2a. Stx2a is considered the most potent Stx subtype and is associated with high virulence and HUS [50,54,55]. As isolate S2 has the potential to produce the more potent Stx2a form of Stx, it is likely to be more virulent than the other three isolates described in this study. All four isolates produced Stx, and the production was increased in the presence of MMC. In a study by Muniesa et al. (2004), 18% of 168 *stx2*-carrying STEC strains, isolated from cattle, were MMC-inducible [56]. Our results indicate a higher production of Stx1 by isolates S1, S2, and S4 compared to the degree of Stx2 production by isolate S2. The kit used for the detection of Stx, the enzyme immunoassay RIDASCREEN^®^ Verotoxin kit (R-biopharm, Darmstadt, Germany), detects all known Stx-types [57]; however, a direct comparison between the amount of Stx1 and Stx2 produced is not applicable as the RIDASCREEN^®^ Verotoxin kit has a lower detection limit for Stx1 (12.5 pg/mL) than for Stx2 (25 pg/mL). The degree of Stx production was examined at 37 °C, as this is the temperature in the human gut where the toxin’s production occurs.

Stx-encoding prophages are very diverse and recent studies have suggested that their pathogenic potential is determined by the phage replication region, encoding the phage repressor protein CI and the phage replication proteins [10,11]. The EHEC phage replication unit Eru1, which is carried by the highly pathogenic EHEC strains that caused the Norwegian O103:H25 outbreak in 2006 and the large O104:H4 outbreak in Europe in 2011, is also carried by the S2 isolate described in this study [10]. Eru1 is often carried together with a Clade II CI repressor, as is the case for the S2 isolate, and may also indicate a high pathogenic potential [11]. It has previously been suggested that phage production is not induced by MMC in the Eru1 type of Stx-phages [10]. Contrary to this suggestion, we show herein that Stx production is induced by MMC in isolate S2, which suggests that Stx production and the production of new phage particles are regulated differently even in phages belonging to the same Eru type.

All four STEC isolates characterized in this study carry the gene encoding intimin, which has been associated with an increased ability to cause severe disease [55]. They also carry the large O157 plasmid harboring the virulence gene *ehxA*, encoding enterohaemolysin, which is present in most isolates from clinical STEC-infections [55]. The gene *astA*, encoding the heat-stable EAST1 toxin, which is present in several human diarrheagenic *E. coli* pathotypes was also found in in the genomes of the four isolates [46]. An EAST1-positive *E. coli* strain has been suggested to be the culprit of a large diarrhea outbreak in Japan that affected 2697 children [58]. The *astA* gene is, however, also commonly found among *E. coli* isolates collected from the environment [59]. The presence of genes encoding Stx, intimin, and enterohaemolysin as well as the EAST1 toxin in *E. coli* isolates from Norwegian dairy farms strongly indicate that Norwegian raw milk may contain highly pathogenic *E. coli*.

As raw milk may contain highly pathogenic bacteria such as STEC, *Listeria monocytogenes*, *Campylobacter,* and *Salmonella*, the temperature used for its storage is critical. In this study, we observed that at 4 °C the STEC levels slightly decreased over 72 h; however, only the reduction of S3 was significant (*p* < 0.01). At 6 °C, there was a trend towards decreased STEC levels over the first 24 h of storage, whereafter the levels were constant over the next 48 h. At 8 °C, there was an increasing trend in the STEC levels. Due to the large variation between the three biological replicates in the growth experiments, the results are not conclusive. However, at each temperature, at least one isolate showed a clear increase or decrease in CFU (*p* ≤ 0.05), indicating that temperatures between 6 and 8 °C for more than 24 h may allow STEC to multiply in raw milk. These results are comparable to previous studies that have shown that *E. coli* O157:H7 is capable of growing in raw milk at 7 and 15 °C [60], but not at 5 °C [31]. Another study showed that *E. coli* O157:H7 did not decrease during storage at 4 °C for five days. However, the study used streptomycin-resistant strains and raw milk supplemented with streptomycin, which may have influenced the natural microbiota of the raw milk [28]. The large growth variations between replicates of the same isolate in our study indicate that even though the growth is not statistically significant, sudden multiplications of STEC can occur in individual milk samples. The experimental conditions in the present study are not directly comparable to natural conditions since the raw milk was inoculated with 3000–5000 CFU/mL and such a high number of STEC is not likely to be present in fresh bulk tank milk. The transition from LB media at 37 °C—used for pre-culturing the isolates—to raw milk at low temperatures may also have influenced the survival of the isolates.

To mimic a real-life scenario of temperature abuse during meals, the milk was stored at 4 °C interrupted by exposure to room temperature (20 °C) for 1.5 h per day. Under these conditions, a general increase in CFU/mL milk was observed after 72 h; however, the increase was only significant for isolate S2 (*p* < 0.05). The recordings of the temperature in 40 mL of raw milk moved from 4 °C to 20 °C showed that the sample reached room temperature after 1.5 h. In a real-life situation, we assume that the volumes of raw milk stored for consumption are larger than 40 mL and the temperature fluctuation in the stored milk will be less pronounced.

Isolate S2 showed rapid growth during the storage of UPM at 20 °C, while the growth rates of the clonal isolates S1, S3, and S4 were slower. However, in LB media, all isolates showed similar growth rates and reached higher concentrations than they did in UPM, stored for the same time. The growth inhibition of the three clonal isolates may be due to the presence of milk-borne antimicrobial components such as lactoperoxidase, lysozyme, xanthine, oxidase, lactoferrin, immunoglobulins, and bacteriocins, or by competing microorganisms [61]. Previous studies have shown a better survival of *E. coli* inoculated in pasteurized milk compared to *E. coli* inoculated in UPM [31]. This is not surprising, since UPM contains an indigenous microbiota that can influence the growth of STEC. Notably, *E. coli* O157:H7 has been shown to be unresponsive to the antimicrobial activity of the lactoperoxidase–thiocyanate–hydrogen peroxide system (LPS) in milk, and this may also be the case for isolate S2 [60].

The survival and growth levels were only examined over a period of 72 h post-inoculation, as raw milk is not recommended to be stored for a very long time before consumption [62]. However, temperature abuse in consumers’ handling practices is common both during transport and storage. Most consumers are unaware of their refrigerator’s temperature [63], and studies show that household refrigerators often hold higher temperatures than recommended. Furthermore, milk is often kept at locations in the refrigerator where the temperature varies, for example, in refrigerator door racks [63,64,65,66]. This is particularly important to consider regarding the risk of disease from low-dose pathogens such as EHEC [67].

## 5. Conclusions

STEC isolates harboring genes associated with pathogenicity such as *stx1/2*, *eae*, *ehxA*, and *astA* are present in Norwegian dairy farms, and potentially pathogenic STEC isolates are able to can grow in raw milk stored at temperatures above 6 °C. As previous studies show that domestic refrigerators often hold higher temperatures than recommended, the growth of STEC in stored raw milk is a likely scenario. Altogether, the results suggest that UPM from Norwegian dairy farms may contain highly pathogenic STEC strains, and that the storage of UPM under suboptimal refrigeration conditions increases the risk for hemorrhagic colitis and HUS. To reduce the risk associated with the consumption of UPM, consumers need more knowledge regarding the importance of keeping the milk sufficiently chilled to prevent the growth and survival of STEC and other pathogenic bacteria. They should also be made aware of that even correctly stored UPM is associated with an increased risk for illness and that young children, elderly, and immunocompromised individuals are particularly vulnerable.

## Figures and Tables

**Figure 1 foods-11-03411-f001:**
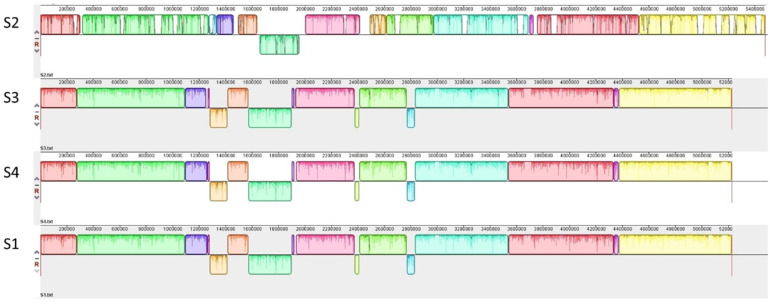
Multiple genome alignment was performed using the Mauve software. Each sequence is represented by a horizontal panel of blocks. The colored blocks indicate homologous sequence regions between the genomes. Blocks below the center line in each genome are inverted sequences with respect to the other genomes.

**Figure 2 foods-11-03411-f002:**
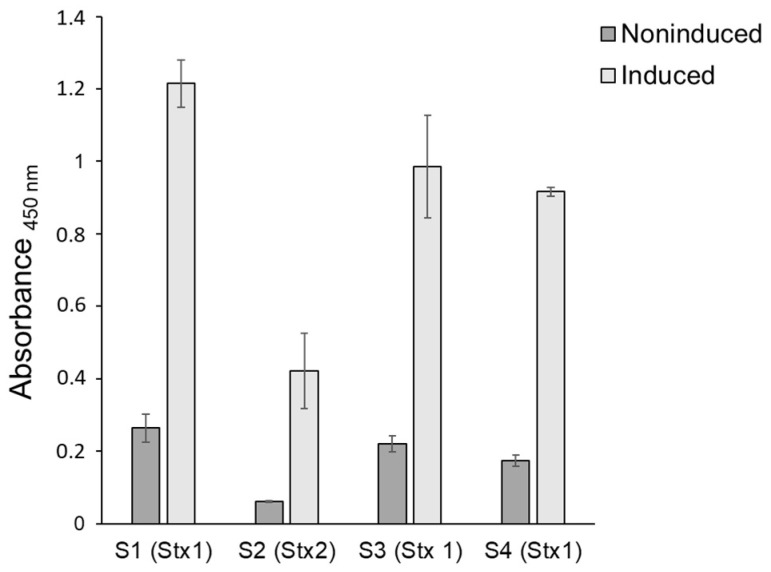
Semi-quantitative determination of Stx production of STEC isolates S1, S2, S3, and S4 after three hours induction with Mitomycin C (0.5 µg/mL). Error bars represent standard deviation.

**Figure 3 foods-11-03411-f003:**
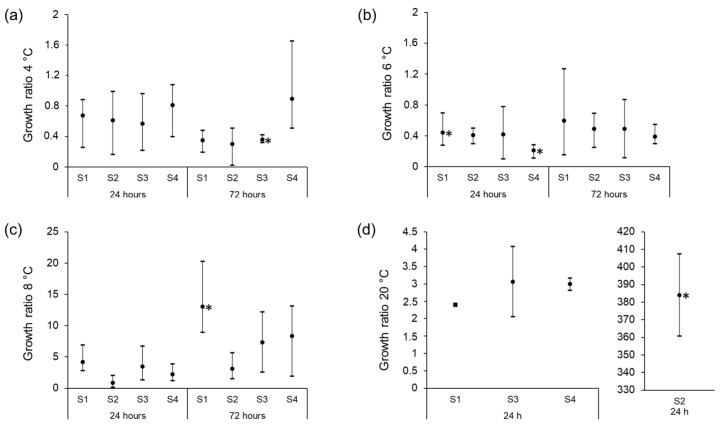
Chart showing the minimum, maximum, and average of growth ratios for STEC isolates S1, S2, S3, and S4 in unpasteurized milk at 4 °C (**a**), 6 °C (**b**), 8 °C (**c**), and 20 °C (**d**). Growth ratios below 1 indicate cell death while a growth ratio above 1 indicates growth. Asterisks represent statistical differences from pairwise comparisons between inoculation point and 24 or 72 h using two-tailed paired Student’s *t* tests (* *p* ≤ 0.05).

**Figure 4 foods-11-03411-f004:**
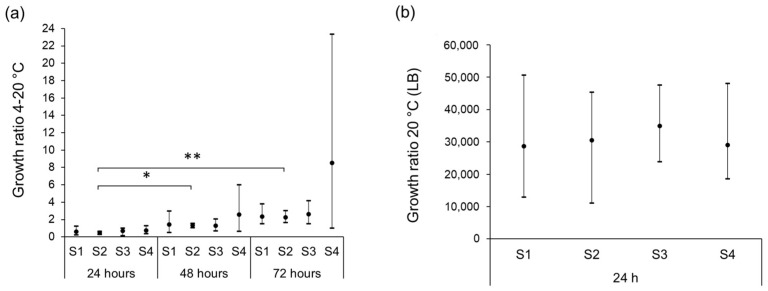
Chart showing the minimum, maximum, and average of growth ratios for STEC isolates S1, S2, S3, and S4 in unpasteurized milk at 4 °C under a temperature abuse scheme of 90 min at 20 °C every 24 h (**a**) and at 20 °C in LB-broth (**b**). Asterisks represent statistical differences from pairwise comparisons determined using two-tailed paired Student’s *t* tests (* *p* < 0.01; ** *p* < 0.05).

**Figure 5 foods-11-03411-f005:**
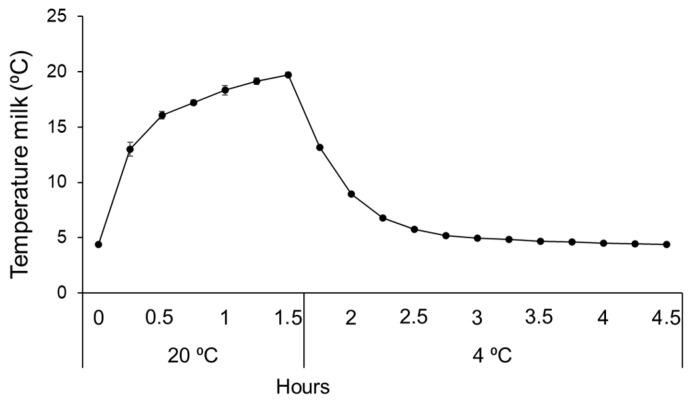
Temperature fluctuation in 40 mL UPM incubated at 4 °C, interrupted with incubation at 20 °C for 90 min. Error bars represent standard deviation.

**Table 1 foods-11-03411-t001:** Characteristics of the Shiga toxin-producing *E. coli* isolated from dairy farms located in the southeast of Norway [25].

	S1	S2	S3	S4
Source	Cattle feces (Farm B)	Cattle feces (Farm A)	Milk filter (Farm A)	Cattle feces (Farm A)
Year of isolation	2019 (November)	2020 (January)	2020 (June)	2020 (June)
Country	Norway	Norway	Norway	Norway
Pathotype	STEC	STEC	STEC	STEC
Serotype	ONT:H28	O108:H25	ONT:H28	ONT:H28
NCBI accession no	JANWGF000000000	JANWGE000000000	JANWGD000000000	JANWGC000000000
LEE operons	five	five	five	five
Intimin type	gamma	alpha	gamma	gamma
*ehxA*	yes	yes	yes	yes
*astA* ST toxin	yes	yes (2)	yes	yes
Stx type	Stx1a	Stx2a	Stx1a	Stx1a
Eru type	lambdoid	Eru1	lambdoid	lambdoid
Stx phage CI clade	V	II	V	V

## Data Availability

The data presented in this study are available on request from the corresponding author.

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
