# Peer review of "The Ability of Shiga Toxin-Producing Escherichia coli to Grow in Raw Cow’s Milk Stored at Low Temperatures"

_foods, 2022, doi:10.3390/foods11213411_

Round 1
Reviewer 1 Report
The manuscript submitted for review presents interesting information related to the presence of toxins in unpasteurized cow milk produced by E.coli, and in particular of genetic characterization of STEC isolates from raw milk. I have a few comments for this:
1. Please describe the statistics in detail in the "Materials and Methods" section (with what program, what test was used, etc.)
2. Please provide an exact description of "raw milk" - what breed, how many samples were taken (how many ml) in the "Materials and Methods" section
3. List which pathogens are it - in Line 328.
4. Please give specific numerical values (temp.) in conclusion.
5. Please provide a conclusion that suggests a possible solution to the problem or indicates the need for further research in "Conclusion".
Reviewer 2 Report
The manuscript “The ability of Shiga toxin-producing Escherichia coli to grow in raw cow´s milk stored at low temperatures” (foods-1982604) is meaningful. However, there are some questions that the authors should clarify to improve the quality of the work.
1. Line 14-15: I think the sentence “Pasteurization is therefore important for protecting consumers from milk-borne infections.” is not suitable for “Abstract”.
2. Line 24: The keywords need to be modified. The “Shiga toxin-producing Escherichia coli” and “STEC” are repeated.
3. Line 38: Please clarify the infectious dose of EHEC.
4. The absolute quantitative virulence gene expression of the four STECs at different temperatures needs to be investigated to more accurately assess the risk of STECs in raw cow´s milk.
5. Line14 and Line 363: I am confused whether pasteurization prevents the growth of the STECs.
Round 2
Reviewer 2 Report
The paper can be accepted.